# Experiences of Bereaved Families by Suicide in South Korea: A Phenomenological Study

**DOI:** 10.3390/ijerph19052969

**Published:** 2022-03-03

**Authors:** Eunjoo Lee

**Affiliations:** Department of Nursing, Kyungnam University, Changwon 51767, Korea; abigail@kyungnam.ac.kr; Tel.: +82-55-249-2424

**Keywords:** suicide, suicide bereavement, qualitative research, experience

## Abstract

When an individual commits suicide, family members frequently experience feeling of hurt, denial, shock and anger, resentment, shame, and guilt. These negative emotions experienced by family survivors make their life suffer and impede the grieving process. If left untreated, they become physically and psychologically vulnerable and the risk of suicidal ideation is high, so professional intervention is needed. This study aimed to explore the experiences of suicidally bereaved families in South Korea. This research was designed to a qualitative phenomenological study conducted by using Colaizzi’s methods. Participants were seven individuals who had lost a family member to suicide in Changwon City, South Korea. The Data were collected through in-depth and individual interviews with participants from June to December 2018, and the interviews took place 1 to 15 months after their loss. A total of 25 meaning units, 12 themes, and 5 theme clusters emerged from the analysis. The 5 themes of south Korean bereaved families’ experience were: (1) an absurd breakup that came without notice, (2) a life trapped in pain, (3) family isolation by themselves, (4) uncontrolled mind in daily life, and (5) liberating from the bondage of pain. The findings of this study provide insight regarding how suicide loss affects bereaved families and could inform the development of evidence-based programs to prevention suicide thought experienced by bereaved families.

## 1. Introduction

Suicide occurs at any time of the life cycle and is a leading cause of death worldwide [1]. The suicide mortality rate in Korea (per 100,000 population) was 25.6 in 2016, which has been gradually decreasing since 26.9 in 2019 [2]. Compared to countries of the Organization for Economic Co-operation and Development (OECD), Korea is the country with the highest suicide rate [3].

The causes of suicide are varied, including psychiatric symptoms, interpersonal relationships, money loss, physical illness, loneliness, work-related problems, chronic poverty, and assault [4]. When a family member dies by suicide, it negatively affects the remaining family’s life for a long time. Young people can experience deep emotional wounds and anger due to the death of their family and may attempt suicide, thus, suicide in such a family environment is recognized as a solution to the problem [5]. Especially, suicidal behavior occurs when a person who commits suicide fails to resolve problems with their family; thus, family suicide has a significant influence on the life of the surviving family after the suicide [6].

Due to suicide, families bereaved by suicide experienced hurt, betrayal, shock, anger, resentment, shame, and guilt [7]. Families who have experienced a death by teenage suicide are severely emotionally disturbed and socially isolated for a long time due to bereavement, grief and loss. There is a lack of social and emotional support to those left behind after the suicide of a family [8]. If suicide-bereaved family members are left unattended, they become more physically and psychologically vulnerable and tend to commit suicide [9].

There are no significant differences in depression, anxiety, and suicidal behavior of the suicide-bereaved family whether the cause of the family member’s death was suicide or natural causes. Nevertheless, family bereaved by suicide have been reported to experience higher levels of rejection, shame, stigma, criticism [10], and complex grief for not preventing the death [9]. They also want to conceal a suicide event of family because of the prejudice against suicide [6]; thus, their mental contraction has made the application of nursing care or therapeutic approaches difficult. Intervention programs for the family bereaved by suicide will require differentiated strategies that reflect their unique characteristics. For this purpose, it is necessary to deeply understand the grief related to loss of the bereaved families by suicide.

Foreign studies related to suicide bereavement included social stigma [6], mental health and sadness reactions [10], complex sadness and suicidal thoughts [9], and internet support group intervention [11]. Various research designs with research methods and variables were used. It found that only a few Korean studies [12,13,14] had investigated the family after suicide bereavement. The studies related to counseling or family therapy interventions focused on mourning their grief and no nursing approach examined the total experience of the family after suicide bereavement. Phenomenological research reveals the meaning of the experience and explores the nature of phenomena through the analysis of participants’ vivid experiences. Therefore, this study aimed to explore the experiences of the bereaved families by suicide by phenomenological research. This study may help to gain a deeper contextual understanding of family bereaved with a view to exploring their nursing needs.

## 2. Materials and Methods

### 2.1. Participants

Research participants were recruited through support program for families after suicide bereavement identified by the K Health Center in Changwon, South Korea. They were adults over 20 years who voluntarily agreed to participate in the study with understanding its purpose and procedure. In Korea, people over 20 years of age are eligible for voluntary research consent because they are adults.

The public health center conducts one-on-one face-to-face counseling with the bereaved families by suicide to prevent suicide and detect high-risk groups of suicide. They were introduced by the public health center staff in charge of the bereaved families by suicide, and none of them refused to participate in the study or gave up halfway through.

In qualitative research, two sampling principles are applied: appropriateness and sufficiency of sampling. Appropriateness refers to the participant who can provide the best information for the study, and sufficiency is the collection of data to reach saturation in order to provide a sufficient and rich explanation of the research phenomenon [15]. In this study, without limiting the characteristics of the participants such as relationship with the suicide family, age, and bereavement period, 7 participants who could fully express the experiences of the bereaved families by suicide were selected

The average age of the participants was 53.7 years (age range, 42–65), and 4 participants (57%) were female and 3 (43%) participants were male. The participants were in paid employment (*n* = 5, 71.4%), retired (*n* = 1, 14.3%), or unemployed (*n* = 1, 14.3%). Mean time after bereavement was 5.3 months (range: 1–15 months). 4 participants were Buddhist while 3 participants did not have a religion. The participants were mother (*n* = 4, 57%), father (*n* = 1, 14%), and husband (*n* = 2, 29%) of the deceased (Table 1).

The researcher himself is the most important research tool for qualitative research [15]. As a nursing major, this researcher naturally became interested in depression and suicidal thought while empathizing with the pain of the bereaved families by suicide. I had the opportunity to observe the program by visiting the treatment of the community-based treatment at a mental health center. I have experience in conducting qualitative research and has published several manuscripts in peer-reviewed journals. I also have attended qualitative research classes and seminars on phenomenology and grounded theory every year and read dozens of books and papers related to qualitative research in order to increase my knowledge of it.

### 2.2. Data Collection

Data were collected through in-depth interviews for approximately six months from June to December 2018. In order to build the credibility of a researcher credibility at the beginning of the interview and to encourage the participants to speak naturally, the first questions were: “How are you doing these days?”. Then, they were asked to open-ended questions. By narrowing the scope to specific questions such as “Please tell us about your feelings regarding the family suicide?”, “How is it to live as a bereaved family?”, participants were encouraged to voluntarily tell their own stories. When the responses were unclear in meaning, several follow up questions were asked until the responses were clarified. When no further new content emerged from the interviews and when concepts and categories of the same type appeared repeatedly, no further data were collected.

The number of interviews was once per participant, and if additional interviews were needed, additional data were collected by using one or two telephone calls. One interview was conducted with each participant and if additional interviews were needed, additional data were collected through one or two telephone calls.

The duration of interviews was 90 and 120 min and were conducted in the interview room of the public health center—a place where there was no interference from others and that had a free and comfortable environment. The interviews were recorded with the consent of the participants. The audio-recordings were transcribed verbatim (Korean language); additional concepts and contents that were not clear were also confirmed. Finally, the participants were asked to check the analysis contents. After the interviews were completed, a reward was provided to the participants. Participants were compensated with a 30,000 Korean won gift certificate.

### 2.3. Data Analysis

Data were analyzed using seven steps from the Colaizzi’s descriptive phenomenological method [16]. Colaizzi’s analysis method [16] is appropriate for phenomenological qualitative research methods, which analyze the experiences of the bereaved families by suicide to reveal the meaning of the experiences and understand them in depth. Therefore, this study described the experiences of bereaved people after family suicide by applying Colaizzi’s phenomenological method. First, I recalled the expressions, appeals, and situations of all participants, and tried to decipher the overall feeling and meaning of the contents by repeatedly reading the transcribed contents of the interview. Second, I independently extracted meaningful statements related to the phenomenon. Subsequently, I constructed a meaningful statement by underlining the contents that were commonly emphasized in the statement and the sections judged to be essential. Third, while considering the meaning of the extracted sentences and phrases, the hidden meaning in the context was found and restated or summarized. Fourth, the meaning was constructed through a process where I examined the validity of the meaningful statement and ensured that the meaning derived from the re-statement did not deviate from the original data. Fifth, the composed meanings were grouped and organized into theme clusters, and theme clusters were regrouped and derived as themes. Sixth, participants’ experiences were categorized as a sub-topic, subject, and collection of subjects according to the subject, and the essential structure was constructed by integrating common elements of the investigated phenomenon. Finally, in order to secure the validity of the analysis results, the researcher showed the described contents and the phenomenological analysis results to the research participants and confirmed whether they agreed.

The rigor of the study was secured by applying the criteria suggested by Lincoln and Guba [17] to verify the quality of the study and increase its validity and reliability. First, the researcher used an audio recorder during the interview to ensure credibility, and analyzed whether he/she correctly understood the content stated during the interviews. In addition, after all the interviews were completed, the interview contents were summarized, and the participants went through a procedure to confirm that the summarized statements matched their experiences. Moreover, the interview contents were transcribed immediately after the interview, and efforts were made to organize the entire process of the interview accurately. The analyzed results were checked by nursing professors and linguists with extensive experience in qualitative research and interviews, who then provided feedback on them. Furthermore, for transferability, interviews were conducted until the research data were saturated, and additional interviews were conducted for areas that were not investigated adequately, to find common experiences. In order to secure auditability, all interviews, data analysis, and analysis tables of the coding process, journals, and memos were kept as records so that they could be cross-checked, if necessary. Finally, confirmability increases the objectivity of data by maintaining the standards of reliability, suitability, and auditability, and does not manipulate any involvement or intentional situations when participants talk about their experiences.

### 2.4. Ethical Consolidation

This study was approved by the Institutional Review Board of Kyungnam University. Before conducting the interview, the researcher explained the purpose and procedure of this study, and recording of the interview to the participants. All participants provided written consent to participate in this study.

## 3. Results

In this study, a total of 25 meaning units, 12 themes, and 5 theme clusters were derived from the analysis (Table 2).

### 3.1. Theme Cluster 1. An Absurd Breakup That Came without Notice

The sudden suicide of the family drove the ordinary life of the bereaved family into shock and anger overnight. The relationship between spouses or parents, which could not be separated from the bereaved, was destroyed in an instant, and their life with the deceased was blown away. They knew they had to accept it, but they could not admit to becoming the bereaved families by suicide, and could not find reasons for the family’s suicide.

#### 3.1.1. Theme 1. Unacceptable Reality

The fact that a loved one suddenly finished his life by committing suicide destroyed the lives of other families. The great tragedy threw them all into a mess and caused anger in their mind. The shock left them with psychological distress and maladaptive physical symptoms. They felt that they could not live rationally as before the incident.


*My child died. I have strange thoughts, I can’t sleep… I was shaking again today. I also experienced incontinence and couldn’t take care of myself today. I think it’s because my body is so shocked. In other words, I cannot believe and accept what has happened.” (Gukjin).*



*“If there is a god, I’d rather get sick or have some kind of an accident and break my leg. This is an insurmountable problem, so I feel a lot of anger. I want to my lives recklessly.” (Eunjeong).*


#### 3.1.2. Theme 2. Anger and Sadness for the Deceased

Participants considered it selfish to choose suicide without knowing that the deceased was loved by his family. The deceased commits suicide to escape the pain, but the bereaved families must live in greater pain. They did not understand that the deceased had chosen to commit suicide to solve the problem on his own without asking for help from his family.


*“If my child thought of us, he wouldn’t have done that. I thought our baby was selfish. Honestly, I tried countless times to die. I didn’t commit suicide because of my family. My son just let go of everything. I’m so angry about that. It’s so sad that he didn’t know that we loved him so much.” (Mija).*



*“Is it true that there is a Buddha? Bad people live well, so why goes Buddha me an ordeal that I can’t overcome?” (Gukjin)*


### 3.2. Theme Cluster 2. A Life Trapped in Pain

The suicide of a family member plunged the rest of the family’s daily life into hell-like pain. The bereaved families summoned the memories with the deceased and held them, and could not get out of the sadness of the loss. They blamed themselves with regret and guilt that they failed to prevent the deceased from committing suicide because they did not recognize the signs of the deceased’s pain.

#### 3.2.1. Theme 1. A Longing That Can’t Be Pushed Away

Participants suddenly recalled the memories they had with the deceased during their lives and were obsessed with these memories. They did not consciously recall the deceased, but when alone, they soon missed the deceased and returned to their state of pain again. They were heartbroken by the fact that they could no longer see the deceased, hear their voice, or smell their odor; this was the most painful experience for the bereaved family members.


*“In the past, my son took us to a famous soup restaurant. My son can’t take me anywhere. If there’s anything I don’t know, my son taught me everything. I didn’t know how to use a cell phone. He was better than a teacher for me.” (Sungmi).*



*“When I enter the house, everything returns to reality. Even if I go outside and have fun while talking and having a cup of tea, it’s too futile when I come home.” (Jiwoo).*



*“I miss my son so much. The hardest thing is that I can’t see my son again in my life. I still sometimes think that he will come out of his room. It seems like he will come back from the academy and say, I’m home.” (Eunjeong).*


#### 3.2.2. Theme 2. Overwhelming Feeling of Loss

When the participants were overcome by the thought that the deceased had disappeared from the world, they could not contain their emotions of extreme sadness. They missed the deceased even more because it seemed that the deceased would appear in the space where they were with him.


*“I just miss my baby so much every day. The hardest part is that I won’t be able to see my child for the rest of my life. The child will still come out of the room and will say, “I’ve been there.” It is the most difficult right now.” (Eunjeong)*



*“What I want to see is the hardest. On Saturdays and Sundays, the children come home. I think I’ll leave the door open on Saturday. I think I’m going to have to sell my house and move out.” (Gukjin)*


#### 3.2.3. Theme 3. Self-Blame Caught in Guilt

Participants remembered the time they spent with the deceased, regretting that they were not able to prevent suicide. Before the suicide, the deceased behaved differently than usual and exhibited warning signs of suicide, but they confessed that they could not stop it because they were not aware that it indicated suicidal behavior. Families continued to recall actions they thought were indifferent toward the victims.


*“My wife had been lying at home for a month before her death. She said she had no appetite. I should have taken better care of her. She said she wanted to take a break, and said it would get better after the break. I didn’t pay much attention to her. Now I knew she was having a hard time, but I didn’t know she would do that. Why didn’t I know it? I regret my behavior.” (Kyungsu).*


### 3.3. Theme Cluster 3. Family Isolation by Themselves

The bereaved family thought that the only relationship in the world where they could share their sorrow was family. People tried to comfort the bereaved family, but they could not share the pain of loss. They were isolated from society because of the shame of being the bereaved family by suicide.

#### 3.3.1. Theme 1. Closed Mind by Unwanted Consolation

Participants were hurt by acquaintances who deprecated suicide, saying that the surviving families do not need to waste money and time at the funeral, and it is shameful to report a suicide in their surroundings. They were displeased with the pitiful attitude of others, who acted as if something unspeakable had happened. Far from consoling the desperate family members, acquaintances treated the death of the deceased as a bystander; this was embarrassing and stunning for the surviving family members.


*“An acquaintance living in Namhae told us that there is no need for a ritual such as a funeral. But when his son committed suicide, he properly arranged the funeral. I heard that he had paid 100 million won to the temple and worshiped the Buddha. He told us not to waste money on mourning the dead, and everything was useless.” (Mija).*



*“After a long time, I met someone, and she looked at me with a pity. It was more hurtful than comforting. When someone’s parents die, people don’t overreact. But when my child died, she asked me ‘are you okay?’. It made me more uncomfortable. So, I just don’t want to contact people unless they are very close. Such attitude is uncomfortable. The death of a parent or a child, it can all happen. We may experience both. I can’t stand people reacting specifically to my son’s death. It’s confusing when I think of myself as someone who should be comforted like this.” (Jiwoo).*



*“After my wife died, I thought that relatives or acquaintances would unilaterally comfort the remaining family members. Some people told me how hard it must have been for my wife to commit suicide. Those words hurt us while we were depressed.” (Kyungsu).*


#### 3.3.2. Theme 2. A Secret to Hide from the World

Participants were conscious of the negative evaluation of suicide in Korean society and feared that they would be known as bereaved families. They could not cry openly and had to hide their feelings because they could not prevent the suicide of their closest family member and felt ashamed that they were living their lives after the death of the deceased.


*“It was rumored that my son’s friend informed the school about the suicide. Other than that, no one knows. Neighbors living next door don’t even know this. My son became a sinner, and we also became sinners. People may say that our daughter belongs to a bereaved family. In Korea, if someone commits suicide, it hurts their family.” (Mija).*



*“It would be okay if I had died, the older one. I can’t talk to anyone about my child’s death. Is this a good thing? Even now, I don’t cry at home because I’m afraid people will find it out. I cry, but I don’t make a loud noise.” (Gukjin).*



*“I don’t meet anyone.” (Sungmin).*


### 3.4. Theme Cluster 4. Uncontrolled Mind in Daily Life

Due to the loss of their family member, the bereaved family lost hope of living in the present and thought their future had disappeared. They despaired of the irreversible reality and lost the will to live. They came to understand the deceased’s suicide and thought suicide as the solution to escape the painful reality.

#### 3.4.1. Theme 1. A Sense of Defeat Living in a Broken World

The bereaved family thought they had lost the “present” they had with their family and the “future” that they would experience over the next days. They were spending their days meaninglessly in a reality that seemed to be trapped in a tunnel with no end. They came to realize the finiteness and the futility of life, and in their deep sorrow, they could do nothing.


*“What are you going to do by being alive? I have no reason to live without him. Even so, I’m really trying to eat and live again because I have a fixed-term job at this city hall… My son died, but I want to buy and eat valuable ingredients. Even if I didn’t learn anything much, I’ve lived without acting badly to the other people. Why has it happened?” (Gukjin).*



*“There are many times when I’m blank. I also remember making memories with my son a lot, and I keep changing my mind several times a day. I have regrets and resentments, and then I am not just doing anything, day by day. I’m just sad.” (Jiwoo).*


#### 3.4.2. Theme 2. Longing for Suicide

Earlier, the participants thought that suicide was a frightening and terrifying thing that had nothing to do with them but could happen to anyone easily. However, now, the participants think that it could be their own fate. After the death of their family member, they were able to understand the suffering of the deceased, who had no choice but to consider suicide as an escape from suffering as they could not live with the pain. They felt themselves close to committing suicide and experienced thoughts of wanting to die.


*“In the past, when I heard news of celebrities’ suicide, I used to think, ‘Why did they die?’ However, after my wife’s suicide, I can understand how one could commit suicide. I thought my wife must have had a hard time. I was afraid of committing suicide and dying, but after my wife’s death, I thought I could die too.” (Sungmin).*



*“Not long ago, I went to the beach and saw the ocean water overflow. I just wanted to dive and drown. I felt like I was going to die easily. I think of dying a lot these days. Although, actually, I didn’t do that, but I wanted to jump off the 16th floor yesterday. The home owner put a safety net on all windows. Other houses don’t have it, but this house has safety nets installed. So, I thought that I chose this house well.” (Mija).*



*“Even now, I just think I want to die like my son.” (Gukjin).*


### 3.5. Theme Cluster 5. Liberating from the Bondage of Pain

The bereaved families were trying in their own way to overcome the painful reality and return to the past. Although it is impossible to completely return to a life without the deceased, they were able to control their thoughts and emotions and plan a new life separate from the deceased.

#### 3.5.1. Theme 1. Struggle to Escape from the Bondage of Memories with a Loved Family

As the participants recalled the deceased, they were sucked into an uncontrollable abyss of sorrow; hence, they tried to immerse themselves in work or live busy daily lives. They consciously focused on work and stopped thinking about the deceased, trying not to face pain.


*“It’s okay when I work. Only when I work. It did not last long. Also, I think where my son will come from, but I forget it when I work.” (Mija).*



*“No matter how close I am, it is correct to distance myself from other people. For now, I’m only talking with my daughter and my husband. I try to train myself harshly and have a strong heart. When a tough thought comes up, I stop it. I work with my mind focused and don’t want to think about anything else. If I think of my son, I may make mistakes.” (Mija).*



*“I go to the temple whenever I have time. I bow down 108 times when I feel frustrated because I think of my child. There are times when I just feel frustrated and upset. At that time, after I did it, my feelings subsided a bit.” (Eunjeong).*



*“I can’t erase my memories with my child. I pray to the Buddha to be with the child in the next life.” (Sungmi).*


#### 3.5.2. Theme 2. Face to Face with Sorrow

Participants were reluctant to talk with their families because they thought that their families would be hurt by the loss of the deceased; however, they gradually began to talk about the death of the deceased and their feelings. They tried to accept the death of the deceased by seeking confront in memories and photos.


*“These days, I talk to my children. ‘Why did your mother become like this?’ I also talk about the part where I am struggling. In the past, I didn’t tell my kids at all.” (Sungmin).*



*“I want to accept my son’s death with my heart. If I’m sad, I cry, and if I want to see him, I look at his pictures. I am doing what my heart tells me to do. I see pictures if I want to see my son, and I cry if I want to cry. My son died. He went to heaven. I would like to accept that fact as it is. That’s why I want to see him more now.” (Eunjeong).*


#### 3.5.3. Theme 3. Hope Gained by Responsibility for Family

Even though the whole family had suffered a terrible thing like the Blue Army, they were trying to get out of despair while concealing each other’s pain, thinking about the family member who would be more affected. They were anxious about losing another family member, and they tried to take better care of each other. They believed they had the strength to endure the difficult time because of the surviving family, and received comfort and support from each other. They were planning their future careers, holding on to new hopes, and gaining the strength to live again.


*“I’m more concerned about my child’s father than myself. Even that day, my husband sank to the floor. He seems to be okay with his work, but he is hurting a lot now.” (Mija).*



*“My husband suddenly lost 7–8 kgs. I’m very worried about him. He only keeps thinking of our dead son. He says he doesn’t know how to raise our other child. He says he’s doing his best right now, but he doesn’t know what to do. But we still have a second child, so we try to strengthen ourselves.” (Jiwoo).*



*“When my wife was alive, I planned to spend time with her when I retire, but now, I wonder how I will survive my remaining life? I plan to work more. I thought about starting a company based on what I was doing when I retired. When my son graduates, I plan to get my son a job at my company and work with him.” (Sungmin).*


## 4. Discussion

This study aimed to understand the experience of bereaved people after family suicide by applying the phenomenological method. Consequently, 25 meanings, 12 themes, and 5 theme clusters were derived. 

In the first theme, “An absurd breakup that came without notice”, the participants recalled the moment they first heard the news of the deceased’s suicide; they were fiercely angry and denied the reality, thinking it would be an unbearable disaster. The shock experienced by the families of suicide victims leads to anger, which is often characterized by being severe enough to be incomparable to the general loss suffered by the family [7]. In particular, children’s suicide is a violent act for parents, causing unacceptable shock, embarrassment, and fear [13]. Feeling betrayed is a common initial reaction to a family member’s death, and it is a phenomenon that appears the same, although there are differences in time in the early stages after the loss of the family [7]. Family members repeat unanswerable questions about why the deceased committed suicide to understand their motives or beliefs [13]. In this process, they are deeply angered by the thought that the deceased didn’t value their relationship [7,8]. In shock and confusion, the bereaved families at the beginning of the bereavement had no one to call for help; they experienced significant delays in cleaning up the place of death or finding adequate support. In this period of emotional turmoil, practical help and guidance is required by bereaved families for decision-making and support services with high accessibility [18,19].

In the second theme, “A life trapped in pain”, the participants experienced unbearable pain due to the entanglement of feelings such as longing and guilt. People bereaved by suicide experience more difficult and complex emotions along with sadness than non-suicide bereavement [12]. The bereaved family who hears or witnesses a sudden report of the death, despairs of a loss and suffers from the absence of an existence that can only be filled by the deceased [20]. They blame themselves, feeling guilty for failing to rescue the deceased from committing suicide [19]. Bereaved families by suicide are believed to have contributed to the death of the deceased through abuse, neglect, or failure to provide adequate assistance [21]. They condemned themselves, thinking that they could not save him because they had missed the signal of suicide [22]. Guilt is a typical emotion experienced by the family members of suicide victims. It makes it difficult for them to live in reality, and it negatively affects their lives in general [23]. They regard themselves as sinners and act to punish themselves. Regret for their actions can increase guilt and leave deeper scars [23]. Bereaved families will have to accept weaknesses, such as guilt and regret, and evaluate themselves positively. Considering these feeling of family bereaved by suicide, it is necessary to help them to have enough time to mourn the reality and to express their feelings [24].

In the third theme, “Family isolation by themselves”, the participants feared prejudice against suicide; thus, they tended to hide the news of the death, and became passive in interpersonal relationships. The bereaved family members do not want to share the pain with each other, which is related to the desire of not making the other family members suffer from more pain [12]. Previous studies also showed that family members were reluctant to talk about the pain with other family members and felt hurt and angry while talking to family and friends [19]. While it is recognized that interventions can help the bereavement process, it may adversely influence the grieving process for the bereaved families by suicide and limit communication with people [25]. Criticism of neglecting the deceased’s death and the attitude of forcing mourning hurt the bereaved families [12]. They experienced negative gazes against suicide and often internalized the stigma [22]. Social prejudice and shame can mark the family with a scarlet letter, causing them to blame themselves. This can interfere with the mourning process and prevent the proper use of support systems for the families [21]. Bereaved families want close people to acknowledge their lives and suicides of their loved ones. We should deliver true words of comfort to the bereaved family rather than sympathy and expression without understanding. Social bias against suicide should be changed in order for the bereaved family to ask for help from the local community and to live life together with society before the suicide incident.

In the fourth theme, “uncontrolled mind in daily life”, the participants identified with the deceased, felt that death was near, or thought that they could commit suicide like the deceased [12]. Bereaved families accept suicide as a solution to stop suffering, and attempt to understand suicide from the perspective of the deceased [26]. They despaired at the fact that the death of their family was irreversible and that the future they had planned with the deceased had disappeared. Bereaved families could have a sense of great emptiness and loss such as losing their hope in the meaning of life [12]. The bereaved family is physically separated from the deceased but is mentally more closely attached. The loss of a loved one can destroy attachments and lead to suicide among the surviving family members [12]. It has been reported that bereaved families by suicide were at high risk of suicidal thoughts and attempts [27]. This is a period when the risk of suicide events is high; thus, it may be required to monitor carefully and intervene with bereaved families by suicide [13]. A safety system for continuous monitoring should be established to prevent suicide and provide support to them [28]. In particular, since mourning is a process of healing, counselors should support the bereaved families so that they can express and accept their pain and sorrow [12].

In the fifth theme, “Liberating from the bondage of pain”, participants tried to control their negative emotions or sadness by keeping themselves busy in daily life. In Lee and Choi’s study [12], the bereaved families by suicide tried to live separately from the deceased through meditation, yoga, or religious activities. They perceive themselves as having to endure suffering by being busy in life, in order to be separated from the pain of suicide in their family [19]. However, it is not desirable for them to be overactive or to engage in maladaptive behaviors such as excessive drinking [19]. Another the participant recalled memories of the deceased and attempted to reconcile their buried dismal feelings while talking about the deceased. The memories with the deceased who torment them should be renewed with memories of the deceased, not the pain of loss. This makes them turn negative experiences into positive ones, so that they can overcome their pain [20]. In addition, the participants approached new hopes by making plans for the future and had the will to take care of the remaining family members. The bereaved family members try to protect the rest of the family despite suffering, because they fear that one of them may commit suicide like the deceased [12]. The bereaved family members seek a way to comfort each other, strive to play the role of the deceased, oppress the painful feelings, and choose to dedicate and sacrifice their lives for the rest of the family [28]; bereaved families should accept their loss and begin to find meaning and purpose in life, which allows them to grow spiritually [19].

According to Kubler-Ross’s, 5 stages of Grief on death and dying are denial, anger, bargaining, depression and acceptance [29]. In this study, although not depending on the passage of time, these stages could be confirmed. However, the bereaved families by suicide experienced more serious symptoms such as suicidal thoughts, shame, and self-blame. All participants in this research were Buddhist. Among religious people in Korea, Buddhism accounts for 0.6%, and among religious people in Changwon, where this study was conducted, Buddhism account for 28% [30]. In Buddhism, suicide is said to make one have the wisdom to see reality properly and experience the practice of mercy [31]. The acceptance of suicide can be understood in the context of the bodhisattva spirit and practice of throwing away one’s body to overcome the sufferings of life in terms of overcoming and transcending death [32]. Buddhism supports people have lost their families to suicide to overcome the pain of loss through meditation and counseling [33]. Among Buddhists, there were some who depended on the Buddha more after the incident, but there were also those who tried to become stronger on their own. There was no particular difference according to the length of the bereavement, but the pain of loss was more severe in the case of child suicide than in the case of not. There was not a particular time (such as birthdays, anniversaries, or other holidays) when the family members are more likely to think about and commit suicide. They only had strong suicidal thoughts in an environment where they could die.

The bereaved families by suicide often experience insensitive responses and intervention aligned with their needs from agencies [25]. This study is meaningful in that it enables the mental health nurses in charge of the bereaved family by suicide at the public health center to plan services focused on their needs in the care of the recently bereaved and understand their emotional suffering. This study had several limitations. First, participants’ characteristics varied by age, gender, and disease, which makes it difficult to generalize the results to others bereaved by suicide. Second, it is difficult to draw out rich individual experiences because they were sampled from one geographical location.

Future research is need to develop an intervention program for the bereaved families by suicide by reflecting their psychological characteristics shown in this study, and to apply its effectiveness in the clinical field.

## 5. Conclusions

This findings from this research add to some implications to existing knowledge. In Korea, there has been a dearth of research specifically examining the mental health for bereaved families by suicide with qualitative methods from a nursing perspective. Bereaved families by suicide had a sense of guilty, anger, and hopelessness and suicide thought and suffer in silence because of the fear of social prejudice. They were not able to effectively seek services because of suicide bereavement. Attention needs to be given to their negative thoughts and emotions following suicide bereavement by staff in supporting the bereaved families by suicide. Thus, it is imperative to provide psychological and financial support and change organizational policies for bereaved families by suicide.

## Figures and Tables

**Table 1 ijerph-19-02969-t001:** General characteristics of participants.

Fake Names	Age	Sex	Job	Religion	Bereavement Period	Relation to Diseased
Kyungsu	60	Male	Office workers	None	15 months	Husband
Mija	49	Female	Business	Buddhism	2 months	Mother
Sungmin	51	Male	Office workers	None	2 months	Husband
Eunjeong	45	Female	Instructor	Buddhism	7 months	Mother
Jiwoo	42	Female	Office workers	None	1 month	Mother
Sungmi	64	Female	None	Buddhism	5 months	Mother
Gukjin	65	Male	Retirement	Buddhism	5 months	Father

**Table 2 ijerph-19-02969-t002:** Experiences of family members bereaved by suicide.

Theme Cluster	Theme	Meaning Unit
An absurd breakup that came without notice	Unacceptable reality	Abnormal signals from the body
		Deny the death of the deceased
	Anger and sadness for the deceased	A selfish choice that turned a blind eye to family
		An incomprehensible reason for suicide
A life trapped in pain	A longing that can’t be pushed away	The deceased who lives in my memory
		The absence of the deceased that can’t be filled
	Overwhelming feeling of loss	The absence of the deceased that can’t be filled
		Despair of never seeing again
	Self-blame caught in guilt	Blame oneself for failing to stop suicide
		Regret for careless behavior
Family isolation by themselves	Closed mind by unwanted consolation	Words that push condolences
		Blaming family members as a bystander to the deceased’s suicide
		A look of commiseration
	A secret to hide from the world	Prejudice that regards suicide as a crime The shame in being bereaved family by suicide
Uncontrolled mind in daily life	A sense of defeat living in a broken world	Lose the will to succeed
		Have no energy to live
	Longing for suicide	Understand why people commit suicide
		Have a suicide thought
Liberating from the bondage of pain	Struggle to escape from the bondage of memories with a loved family	Live in a rush
		Control one’s memories and emotions
	Face to face with sorrow	Talk about the deceased with family
		Remember memories with the deceased
	Hope gained by responsibility for family	Take care of other members of the family
		Plan the future with family

## Data Availability

The data that support the findings of this study are available on request from the corresponding author.

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
