# Peer review of "Experiences of Bereaved Families by Suicide in South Korea: A Phenomenological Study"

_ijerph, 2022, doi:10.3390/ijerph19052969_

Round 1

Reviewer 1 Report

International Journal of Environmental Research and Public Health (IJERPH)

Review Letter to Authors

Manuscript Title:           Experiences of bereaved families by suicide in South                                              Korea: A phenomenological study

Manuscript ID:             None Provided

Date:                              Saturday, January 22, 2022

_________________________________________________________

This qualitative phenomenological study used Colaizzi’s method to explore the experiences of suicidally bereaved family in South Korea. This research focused on the perspectives of seven individuals who had lost a family member to suicide in Changwon City, South Korea. The Data were collected through in‐depth and individual interviews with 15 participants from June to December 2018, and the interviews took place 1 to 15 months after their loss.

I found this topic quite interesting and the topic very timely given the various emotions that individuals experience when a family member commits suicide. With that said, it is in the spirit of improving the manuscript that I pose the following comments and/or questions:

ABSTRACT

In your Abstract you wrote:

Abstract: When a family commits suicide, family members experience feelings of hurt, denial, shock and anger, resentment, shame, and guilt.

Change to:

Abstract: When an individual commits suicide, family members frequently experience feelings of hurt, denial, shock and anger, resentment, shame, and guilt.

On Page 1 you wrote:

The causes of suicide are various, including psychiatric symptoms, interpersonal relationships, money loss, physical illness, loneliness, work-related problems, chronic poverty, assault, and studies [4]. When a death of a family member is by suicide, it negatively affects the remaining family’s life for a lone time. Young person experiences wounds and anger due to the death of their family or assault, or attempt suicide; suicide in such a family environment is recognized as a solution to the problem [5]. Especially, suicidal behavior occurs when person who commits suicide fails to resolve problems with their family; thus, family suicide has a significant influence on the life of the surviving family after the suicide [6].

Change to:

The causes of suicide are various, including psychiatric symptoms, interpersonal relationships, money loss, physical illness, loneliness, work-related problems, chronic poverty, and assault [4]. When a family member dies by suicide, it negatively affects the remaining family’s life for a long time. Young people can experience deep emotional wounds and anger due to the death of their family and may attempt suicide, thus, suicide in such a family environment is recognized as a solution to the problem [5]. Especially, suicidal behavior occurs when a person who commits suicide fails to resolve problems with their family; thus, family suicide has a significant influence on the life of the surviving family after the suicide [6].

On Page 2 you wrote:

There had no significant difference in depression, anxiety, and suicidal behavior of the suicide-bereaved family whether the cause of family’ death was suicide or natural death.

Change to:

There are no significant difference in depression, anxiety, and suicidal behavior of the suicide-bereaved family whether the cause of the family member’s death was suicide or natural causes.

On Page 2 you wrote:

Colaizzi's analysis method [15] is appropriate for phenomenological qualitative research methods, which analyze the experiences of the bereaved families by suici to reveal the meaning of the experiences and understand them in depth.

Change to:

Colaizzi's analysis method [15] is appropriate for phenomenological qualitative research methods, which analyze the experiences of the bereaved families by suicide to reveal the meaning of the experiences and understand them in depth.

  1. MATERIALS AND METHODS

* Is there a particular reason why you secured participants that were older than 20 years of age?

* You stated that seven participants were selected. How many total did you start with? Did the other potential participants discuss suicide? If so, why were they not included? How did you end up with the seven participants that you did?

* State 4 participants (57%) were female and 3 (43%) participants were male.

* How many participants had children? If so, how many children did they have, on average? What were the ages of their children?

* How did you secure the participants?

* What was the age range of participants?

* What was the average age of participants?

* Mention you had 4 Mothers (57%), 2 Husbands (29%), 1 Father (14%)

* What reward was provided to the participants?

* On Page 3 you mentioned “The researcher used a recorder.” Was an audio or video recorded used?

* Was this the first family member that committed suicide?  

* Was reliability established? If so, what was the reliability percent?

* With such a small sample, I recommend using pseudonyms (fake names) instead of numbers.

On Page 4 you wrote:

Furthermore, for transferability, participants who could sufficiently express their experiences were selected, and interviews were conducted until the research data were saturated, and additional interviews were conducted for areas that were not investigated adequately, to find common experiences.

* How did you determine of a participant “could sufficiently express their experiences?” I am confused by this statement.

* How many additional interviews were conducted? [Related to this, how many total interviews were conducted?]

On Page 4 you wrote:

The fact that a loved one suddenly finished his life by committing suicide destroyed the lives of family survivors in a trice.

* What do you mean by “trice?” This does not make sense.

On Page 4, you wrote:

“My son can take me anywhere.”

Change to:

“My son can’t take me anywhere.”

On Page 5 you wrote:

Participants were conscious of the negative evaluation of suicide in Korean society and feared that they would be known as bereaved families. They could not cry openly and had to hide their feelings because they could not prevent the suicide of their closest family member and felt ashamed that they were living their lives after the death of the deceased.

Change to:

Participants were conscious of the negative evaluation of suicide in Korean society and feared that they would be known as bereaved families. They could not cry openly and had to hide their feelings because they could not prevent the suicide of their closest family member and felt ashamed that they were living their lives after the death of the deceased.

On Page 6 you wrote:

The bereaved family thought they had lost the “present” they had with their family and the “future” that they would experience over the next days. They were spending their days meaninglessly in a reality that seemed to be trapped in a tunnel with no end. They came to realize the finiteness and the futility of life, and in their deep sorrow, they could do nothing.

Change to:

The bereaved family thought they had lost the “present” they had with their family and the “future” that they would experience over the next days. They were spending their days meaninglessly in a reality that seemed to be trapped in a tunnel with no end. They came to realize the finiteness and the futility of life, and in their deep sorrow, they could do nothing.

On Page 6 you wrote:

Earlier, the participants thought that suicide was a frightening and terrifying thing that had nothing to do with them but could happen to anyone easily. However, now, the participants think that it could be their own fate. After the death of their family member, they were able to understand the suffering of the deceased, who had no choice but to consider suicide as an escape from suffering as they could not live with the pain. They felt themselves close to committing suicide and experienced thoughts of wanting to die.

Change to:

Earlier, the participants thought that suicide was a frightening and terrifying thing that had nothing to do with them but could happen to anyone easily. However, now, the participants think that it could be their own fate. After the death of their family member, they were able to understand the suffering of the deceased, who had no choice but to consider suicide as an escape from suffering as they could not live with the pain. They felt themselves close to committing suicide and experienced thoughts of wanting to die.

On Page 6 you wrote:

As the participants recalled the deceased, they were sucked into an uncontrollable abyss of sorrow; hence, they tried to immerse themselves in work or live busy daily lives. They consciously focused on work and stopped thinking about the deceased, trying not to face pain.

Change to:

As the participants recalled the deceased, they were sucked into an uncontrollable abyss of sorrow; hence, they tried to immerse themselves in work or live busy daily lives. They consciously focused on work and stopped thinking about the deceased, trying not to face pain.

On Page 7 you wrote:

Even though the whole family had suffered a terrible thing like the Blue Army, they were trying to get out of despair while concealing each other's pain, thinking about the family member who would be more affected. They were anxious about losing another family member, and they tried to take better care of each other. They believe that they have the strength to endure the difficult time because of the surviving family, and the comfort and support they provided each other. They were

planning their future careers, holding on to new hopes, and gaining the strength to live again.

Change to:

Even though the whole family had suffered a terrible thing like the Blue Army, they were trying to get out of despair while concealing each other's pain, thinking about the family member who would be more affected. They were anxious about losing another family member, and they tried to take better care of each other. They believe that they have the strength to endure the difficult time because of the surviving family, and the comfort and support they provided each other. They were

planning their future careers, holding on to new hopes, and gaining the strength to live again.

On Page 7 you wrote:

“I’m more concerned about child's father than myself…..

Change to:

“I’m more concerned about my child's father than myself…..

On Page 7 you wrote:

Deny the death of the decease

Change to:

Deny the death of the deceased

On Page 8 you wrote:

People Bereaved by suicide experience more difficult and complex emotions along with sadness than non-suicide bereavement [12].

Change to:

People bereaved by suicide experience more difficult and complex emotions along with sadness than non-suicide bereavement [12].

On Page 8 you wrote:

Considering these feeling of family bereaved by suicide, it is necessary to help the them to have enough time to mourn the reality and to express their feelings [24].

Change to:

Considering these feelings of family bereaved by suicide, it is necessary to help them have enough time to mourn the reality and to express their feelings [24].

On Page 9 you wrote:

It has been reported that bereaved families by suicide were at high risk of suicidal attempts and suicide thoughts [26].

Change to:

It has been reported that bereaved families by suicide were at high risk of suicidal thoughts and attempts [26].

On Page 9 you wrote:

Safety system for continuous monitoring should be established to prevent suicide and provide support to them [27].

Change to:

A safety system for continuous monitoring should be established to prevent suicide and provide support to families [27].

On Page 9 you wrote:

Another the participant recalled memories of the deceased and attempted to reconcile with the buried dismal feelings while talking about the deceased.

Change to:

Another participant recalled memories of the deceased and attempted to reconcile their buried dismal feelings while talking about the deceased.

DISCUSSION

  • Provide a Implications for Clinicians section in your Discussion section

  • Provide a Directions for Future Research section in your Discussion section

OTHER ISSUES

  • I found the manuscript difficult to read in parts because the English was so poor. Since the author is not a native English speaker, I recommend that an individual that is fluent in English read the manuscript. The manuscript needs to be edited.
  • On what theoretical framework is the study based? I recommend that you mention Kubler-Ross’s 5 Stages of Grief. Refer to this article:

De Beurs, D., Fried, E. I., Wetherall, K., Cleare, S., O’Connor, D. B., Ferguson, E., ... & O’Connor, R. C. (2019). Exploring the psychology      of suicidal ideation: A theory driven network analysis. Behaviour    Research and Therapy, 120, 103419.

  • You secured interviews from 7 participants: 4 participants were Buddhist while 3 participants did not have a religion. What differences did you find between the responses of the Buddhist participants and the participants that did not have a religion? This must be clear.
  • What differences did you notice between the males and the females?
  • The participants lost their loved one between 1 month and 15 months? As I read the findings, I was not sure if there was a difference in level of shock, anger, frustration, and depression, etc. among the individuals who lost their relative 1 month, 2 months, 2 months, 5 months, 5 months, 7 months, or 15 months later? In other words, how are the responses of individuals interviewed different OR the same based on the months after they lost their family member to suicide?
  • What does Buddhism say about suicide?
  • How prevalent is Buddhism in South Korea?
  • In what ways does Buddhism support individuals that lost family to suicide?
  • In what ways does Buddhism not support individuals that lost family to suicide?
  • Does having a religion (Buddhism) OR not (No religion) make a difference in regard to how well a family can cope when their family member commits suicide?
  • Is there a particular time after the death when family members are more likely to think about and commit suicide? [For example, during birthdays, anniversaries, or other holidays?]

Author Response

Thanks to your review, I was able to write a quality article. Thank you very much.

Reviewer 2 Report

I thank the authors for the opportunity to review this interesting article, it is a interesting article. However, keep in mind the following recommendations and give answer to the questions posed:
• The information on the type of analysis used in the study (lines 64 – 68) should not appear in the introduction, but rather in the corresponding section.
• Justify why they carried out a phenomenological study and not another type of qualitative study.
• In lines 80 and 81, the authors state: “In a qualitative study…six to eight units are required when composed of a homogeneous group; Please provide the bibliographical source that supports this statement. In turn, having 6-8 participants to obtain a homogeneous sample does not mean that it is sufficient when it comes to delimiting the sample size.
• When selecting the participants, the periods of mourning are very different, the loss of a relative after 1 month is not the same as after 15. Most likely, not all the relatives will be in the same phase of mourning. Please explain to what extent it influenced the data collection.
• In relation to the previous point, I am concerned about the ethical implications, in those relatives where the loss was recent. Perhaps broaching the subject too soon could harm the participants. how did they handle this?
• They must write a section on ethical and legal considerations; specify the ethics committee that assessed the study.
• They must rewrite the results. Each of the results must describe the contents of the verbatims. The sustenance of the sections is in the description, not only in the vebatims. Authors must realize this, in some subjects, the verbatims are longer than their description.
• Synthesize the verbatims more, keep in mind that when presenting the results, the sustenance is found in the description
• Do not consider the thematic groups as subsections in themselves, but integrate them within the main topics, it will give more consistency to the writing of the results.
• In line 289, it talks about: “Blue Army” ¿what is it referring to¿. Of more details. Lines 289 – 294 should not be italicized.
• In line 308 they state: Colazzi's phenomenological method. Colazzi is not a type of phenomenological study, but a type of results analysis.

All the best

Author Response

(The authors gave the same response as above.)

Round 2

Reviewer 1 Report

International Journal of Environmental Research and Public Health (IJERPH)

Review Letter to Authors

Manuscript Title:           Experiences of bereaved families by suicide in South                                              Korea: A phenomenological study

Manuscript ID:             None Provided

Date:                              Sunday, February 20, 2022

_________________________________________________________

This qualitative phenomenological study used Colaizzi’s method to explore the experiences of suicidally bereaved family in South Korea. This research focused on the perspectives of seven individuals who had lost a family member to suicide in Changwon City, South Korea. The Data were collected through in‐depth and individual interviews with 15 participants from June to December 2018, and the interviews took place 1 to 15 months after their loss.

I found the revised manuscript much easier to read. Thank you for your time and attention to this matter. With that said, there are still some issues that need to be addressed.

On Page 1 you wrote:

This study aimed to explore the experiences of suicidally bereaved 12 family in South Korea.

Change to:

This study aimed to explore the experiences of suicidally bereaved families in South Korea.

On Page 1 you wrote:

The 5 themes of south Korean bereaved families’ experience were: (1) an absurd breakup that came without notice, (2) a life trapped in pain, (3) family isolation by themselves, (4) uncontrolled mind in daily life, and (5) Liberating from the bondage of pain.

Change to:

On Page 1 you wrote:

The 5 themes of south Korean bereaved families’ experience were: (1) an absurd breakup that came without notice, (2) a life trapped in pain, (3) family isolation by themselves, (4) uncontrolled mind in daily life, and (5) liberating from the bondage of pain.

On Page 1 you wrote:

The findings of this study provide specific insights how suicide loss affects bereaved families and could inform further development of evidence-based programs to prevention suicide thought experienced by bereaved families.

Change to:

The findings of this study provide insight regarding how suicide loss affects bereaved families and could inform the development of evidence-based programs to prevention suicide thought experienced by bereaved families.

On Page 1 you wrote:

The causes of suicide are various,

Change to:

The causes of suicide are varied,

On Page 1 you wrote:

Families who have experienced a death by the teenage suicide are severely emotionally disturbed and socially isolated for a long time due to bereavement, grief and loss.

Change to:

Families who have experienced a death by teenage suicide are severely emotionally disturbed and socially isolated for a long time due to bereavement, grief and loss.

On Page 2 you wrote:

These studies related to counseling or family therapy 62 interventions focusing on mourning their grief; no nursing approach examined the total 63 experience of the family after suicide bereavement.

Change to:

The studies related to counseling or family therapy interventions focused on mourning their grief and no nursing approach examined the total experience of the family after suicide bereavement.

On Page 2 you wrote:

The public health center conducts one-on-one face-to-face counseling with the bereaved families by suicide to prevent suicide and detect stage high-risk groups of suicide.

Change to:

The public health center conducts one-on-one face-to-face counseling with the bereaved families by suicide to prevent suicide and detect high-risk groups of suicide.

On Page 3 you wrote:

The researcher interviewed in this study has many experiences in initial counseling for the bereaved families by suicide and has the ability to cope with psychological crises.

Change to:

The researcher has much experience in initial counseling for bereaved families by suicide and has the ability to cope with psychological crises.

* Not sure what you mean “the ability to cope with psychological crisis.” Be clear.

On Page 3 you wrote:

Another research has experience in conducting qualitative researches and publishing them in academic journals. We attended qualitative research classes and seminars on phenomenology and grounded theory and read books and papers related to qualitative research in order to increase our knowledge of it.

Change to:

Another researcher has experience in conducting qualitative research and has published several manuscripts in peer-reviewed journals. We attended qualitative research classes and seminars on phenomenology and grounded theory and read books and papers related to qualitative research in order to increase our knowledge of it.

* How many classes and seminars have you attended?

* How many books and papers on qualitative research have you read?

On Page 3 you wrote:

By narrowing the scope to specific questions such as, participants were encouraged to voluntarily tell their own stories.

Change to:

By narrowing the scope to specific questions such as __________ [WHAT? Be Clear], participants were encouraged to voluntarily tell their own stories.

On Page 3 you wrote:

The number of interviews was once per participant, and if additional interviews were needed, additional data were collected by using one or two telephone calls.

Change to:

One interview was conducted with each participant and if additional interviews were needed, additional data were collected through one or two telephone calls.

You wrote:

Participants were compensated with a gift certificate.

* What was the amount of the gift certificate? Be clear.

On Page 3 you wrote:

Therefore, this study aimed to descript thoroughly about the phenomenon regarding experiences of bereaved people after family suicide from applying the Colaizzi’s phenomenological method [15].

Change to:

On Page 3 you wrote:

Therefore, this study described the experiences of bereaved people after family suicide by applying Colaizzi’s phenomenological method [15].

On Page 7 you wrote:

Due to the loss of the family, the bereaved family has lost hope of living in the present and thought that their future had also disappeared.

Change to:

Due to the loss of their family member, the bereaved family lost hope of living in the present and thought their future had disappeared.

On Page 7 you wrote:

Although it is impossible to completely return to a life without the deceased, they was able to become the subject of his own life, controll their emotions and thoughts, and planned a new life separated from the deceased.

Change to:

Although it is impossible to completely return to a life without the deceased, they were able to control their thoughts and emotions and plan a new life separate from the deceased.

On Page 8 you wrote:

Also, I think where my won

* Did you mean to write “won?” If so, define “won” for the reader.

On Page 8 you wrote:

They tried to accept the death of the deceased with their hearts, while confronting their memories and photos.

Change to:

They tried to accept the death of the deceased by seeking confront in memories and photos.

On Page 8 you wrote:

They believe that they have the strength to endure the difficult time because of the surviving family, and the comfort and support they provided each other.

Change to:

They believed they had the strength to endure the difficult time because of the surviving family, and received comfort and support from each other.

On Page 11 you wrote:

All participants with religion in this research were Buddhist.

Change to:

All participants in this study were Buddhist.

On Page 11 you wrote:

This study has several limitations. First, participants’ characteristics vary by age, gender, and relation to diseased, which makes it difficult to generalize the results to others bereaved by suicide.

Change to:

This study had several limitations. First, participants’ characteristics varied by age, gender, and disease, which makes it difficult to generalize the results to others bereaved by suicide.

On Page 11, you wrote:

There wasn't a particular time (such as birthdays, anniversaries, or other holidays) when the family members are more likely to think about and commit suicide.

Change to:

There was not a particular time (such as birthdays, anniversaries, or other holidays) when the family members are more likely to think about and commit suicide.

On Page 11 you wrote:

In this study, there were no significant differences between participants with and without religion.

* Delete this statement because you should ONLY use the term “significance” when referring to a quantitative study, or a study that uses numbers.  

* However, what were the differences between participants that were and were no religious?

On Page 12 you wrote:

It is need to provide the boost help such as psychological financial support and changes of organizational policies for them.

Change to:

Thus, it is imperative to provide psychological and financial support and change organizational policies for bereaved families by suicide.

Discussion

* Since your study has already been conducted, your write-up should be in past tense. You need to fix the many tense issues that are in the Discussion section of your paper.

For example, you wrote this on Page 10:

In shock and confusion, the bereaved families at the beginning of the bereavement have no one to call for help; they experience significant delays in cleaning up the place of death or finding adequate support.

Change to:

In shock and confusion, the bereaved families at the beginning of the bereavement had no one to call for help; they experienced significant delays in cleaning up the place of death or finding adequate support.

Examples of Present and Past Tense:

Recall [present tense]

Recalled [past tense]

Have [present tense]

Had [past tense]

Experience [present tense]

Experienced [past tense]

You wrote:

Conclusions

Change to:

Conclusion

Author Response

Thanks to the comments of the reviewer, I was able to write quality article. Thank you very much.

Reviewer 2 Report

After the changes introduced by the authors, the article has substantially improved

Author Response

(The authors gave the same response as above.)
